# Everyday Life after the First Psychiatric Admission: A Portuguese Phenomenological Research

**DOI:** 10.3390/jpm12111938

**Published:** 2022-11-21

**Authors:** Margarida Alexandra Rodrigues Tomás, Maria Teresa dos Santos Rebelo

**Affiliations:** 1Escola Superior de Saúde Atlântica, 2730-036 Barcarena, Portugal; 2Nursing Research, Innovation and Development Centre of Lisbon (CIDNUR), Nursing School of Lisbon (ESEL), 1600-096 Lisbon, Portugal

**Keywords:** psychiatric department, hospital, patient discharge, life change events, psychiatric nursing, qualitative research, phenomenology of practice

## Abstract

Returning to daily life after psychiatric admission can be difficult and complex. We aimed to explore, describe and interpret the lived experience of returning to everyday life after the first psychiatric admission. We designed this research as a qualitative study, using van Manen’s phenomenology of practice. We collected experiential material through phenomenological interviews with 12 participants, from 5 June 2018 to 18 December 2018. From the thematic and hermeneutic analysis, we captured seven themes: (1) (un)veiling the imprint within the self; (2) the haunting memories within the self; (3) from disconnection to the assimilation of the medicated body in the self; (4) from recognition to overcoming the fragility within the self; (5) the relationship with health professionals: from expectation to response; (6) the relationship with others: reformulating the bonds of alterity; (7) the relationship with the world: reconnecting as a sense of self. The results allow us to establish the phenomenon as a difficult, complex, demanding and lengthily transitional event that calls into question the person’s stability and ability for well-being and more-being. Thus, implementing structured transitional interventions by health services seems crucial. Mental health specialist nurses can present a pivotal role in establishing a helping relationship with recovery-oriented goals, coordinating patients’ transitional care, and assuring continuity of care sensitive to the person’s subjective experiences, volitions, and resources.

## 1. Introduction

The meaning of a psychiatric admission entails an undeniable experience of mental distress and the establishment of inpatient treatment which inevitably implies a psychopharmacological approach, whereby patients often express unpleasantness [1].

The literature has recognized the difficulty and complexity of this particular transitional movement, stating the immediate post-discharge period as one with a greater incidence of mortality, self-harm, suicide, violent criminality, aggression, and inpatient readmission [2,3,4]. Several studies describe the hurdles inherent in this period, such as: The symbolic weight of psychiatric admission, which is powerful enough to put in question people’s competence to autonomously manage life, concurrent to social recognition of mental illness through medical diagnosis [5,6,7];The stigma associated with psychiatric hospitalization, which denotes individuals in a derogatory way producing social discredit [8,9,10];The ward’s artificial environment which provides protection, psychological support, and social interaction that, despite appreciated, does not prepare for daily living [2,11];The difficulty in structuring daily activities, facing pre-admission stressors, resuming social roles and responsibilities [2,7,11,12];The expectations established during discharge, which anticipated readiness to return home, but when facing reality, contribute to a sense of disappointment when patients struggle to make it in the community [2,6,13].The perceived view of psychiatric hospitalization can present a more custodial than therapeutic experience [12]. It defines many rules, sometimes restrictive and conducive to a perception of being in a military training camp due to the authority and power of health professionals [5]. Furthermore, it may constitute a traumatic event as patients perceive medications, commitment, police intervention (in transport and sometimes with handcuffs), compulsory treatment, and restraint as punishment or threat [14,15].and, Treatment also comprises a grinding issue. The value of medication comes through as essential for symptomatic suppression. However, side effects (such as sedation or slurred speech) appear to be debilitating, impairing the return to daily activities, namely work or studying [1,13]. Additionally, medication’s purpose sometimes falls short. Patients do not find it helpful to overcome difficulties and may facilitate means to numb and forget problems through self-medication or concurrent use of psychotropic substances [2,12,13,16].

Hitherto, research has provided evidence that returning to everyday life after psychiatric admission is a challenging period of readjustment whereby multidimensional influencing factors shape people’s ability to respond to this transitional event successfully. About this matter, the World Health Organization pointed out care transitions as the person’s movement between different parts of the health care system inherently encompassing a vulnerability in assuring continuity of care, especially when it concerns transition between hospitals and primary care settings [17]. Additionally, it establishes that transition of care goes beyond clinical handover, including subjective views, perspectives and, needs particular to each patient [17]. Hence, returning to daily living after discharge from an inpatient psychiatric admission is a transitional challenge that requires understanding the lived experience of those who endure such an event. 

Regarding this focus of interest, two very recent literature reviews [10,18] emphasize limitations regarding the extent and depth of research highlighting the need to study specific populations, such as: in non-Anglophone countries; in returning to daily life after a first hospitalization; and in a specific age or diagnostic group.

In Portugal, the process of deinstitutionalizing people with mental illness appears in a 2012 study based on the narratives of twenty people who had experienced institutionalization and returned to the community [19]. However, its amplitude does not reveal the lived experience in detail, nor does it focus on the whole movement of returning to daily life, removed from institutional spaces.

Therefore, this work arises from the concern about the person’s adaptation or adjustment after hospital discharge from an acute psychiatry service. In particular, when it occurs for the first time, people have to deal with an unknown event about which they have no lived experience except the preconceived knowledge tacitly ingrained in everyday thinking.

According to the literature, the first hospitalization is a turning point whose impact implies a relevant redirection of life’s trajectory [5,20,21]. The disease becomes part of the public sphere. Formal recognition established by a medical diagnosis implies the need to stay in a hospital. This first hospitalization usually takes place in situations of severe disruption [22], retrospectively understood as “not being successful in the community” [5] (p. 144). In many countries, namely Portugal, inpatient psychiatric admission is the primary therapeutic option for people experiencing acute mental health crises, often associated with the beginning of follow-up and attending mental health services [21]. The person starts to have more structured and hospital-centered care [21]. The usual therapeutic plan will be establishing psychotropic medication with follow-up in an outpatient consultation (often in the hospital) [23]. Only a fringe has access to multidisciplinary care in Portugal, and significant regional asymmetries of community mental health care remain [23,24].

Furthermore, this first hospitalization requires, in a more or less straightforward or complex way, changes that may include: a healthier lifestyle (e.g., sleep hygiene, healthy eating, and avoidance of unprescribed or illegal drugs); medication adherence; management of iatrogenic effects; or recognition of the experience of mental illness. All this is not easy or even considered necessary by some patients. Hence, responding to a recognized knowledge gap through capturing and comprehending the lived meaning of returning to everyday life-after the first psychiatric admission-is paramount [9,10,18]. Particularly regarding the first hospitalization as it did not receive the proper look and attention it needed. 

Health professionals, namely nurses, are particularly attuned to the impact of transitional challenges on people’s ability to cope with such a clear hurdle and resume their daily lives. Understanding the lived meanings embedded in such an occurrence is pivotal to providing informed care and optimizing nursing care, mental health services and policies. Otherwise, it is unavoidable to think that nursing would fall far short of its purpose if it avoided the effort to understand the world of being as being-in-the-world. Thus, this would limit the existential possibilities of human beings to whom nursing is committed to taking care of the “development of human potential, at well-being and more-being” [25] (p. 15).

In light of a humanistic framework, we determined the research question—What is the person’s lived experience of returning to daily life after the first admission to an acute psychiatric service? Concerning the aims of this study we stipulated: To explore, to describe, and to interpret the lived experience of returning to everyday life after the first psychiatric admission.

## 2. Materials and Methods

To access the world of the lived experience of those who returned to everyday life, after their first and only psychiatric admission, the authors rooted this study on the interpretive paradigm, using a qualitative methodology based on van Manen’s phenomenology of practice [26]. In what we consider an eclectic and pragmatical approach to hermeneutic phenomenological research and writing, phenomenology of practice offers procedural steps that guide investigators in gathering, reflecting, and writing on and about the taken-for-granted practices that sustain daily living. Van Manen’s method seeks to produce and enhance action-sensitive knowledge of everyday living embedded in professional, personal, and social practices [26]. Consequently, “phenomenology of practice distinguishes itself from the more purely philosophical phenomenologies that deal with theoretical and technical philosophical issues” [26] (p. 212). 

The method outlined here involves practical and reflective activities that presuppose a specific order, which does not mean proceeding rigidly and strictly. Van Manen suggests tailoring the research method to the phenomenological topic under study as it is possible to work on several activities intermittently or simultaneously and create methods that allow us to access the phenomenon better [26,27]. The main orientations provided consist of philosophical, philological, and human science methods. These cross the entire investigative process, and the researcher fluidly mobilizes them to outline an investigation that makes sense and best serves its purpose.

Therefore, we established as guiding steps for developing this phenomenological study the following:Formulation of the phenomenological question;Gathering of experiential material;Thematic and hermeneutic analysis;Writing of the phenomenological text.

Furthermore, throughout this research, a phenomenological attitude endured via discipline and constant awareness of the researchers. That meant applying an unprejudiced opening of oneself to the primordial immanent experience of the lifeworld where the *epoché* puts everything that results from the natural attitude in suspension so that we can contact the world, or the things of the world, as they show themselves, in and from themselves [26], i.e., “To the things themselves!” [28] (p. 50). Consequently, we had to recognize and integrate two reciprocal movements: the epoché-reduction as a suspension of taken-for-granted knowledge and the reduction-proper as a reflection that seeks experiential radicality through reflection on a phenomenon’s uniqueness that shows or gives itself in its singularity [26].

### 2.1. Formulation of the Phenomenological Question

In phenomenology, the question focuses on the lived meaning of the human phenomenon that is experientially recognizable, accessible, and wholly explored in the research [26]. Our commitment began and materialized with the following question: What is the person’s lived experience of returning to daily life after the first admission to an acute psychiatric service?

To maintain a robust and phenomenon-oriented connection, one must be constantly aware of this issue [27]. Writing down our assumptions and pre-understandings became essential to preserve a genuine openness to the phenomenon. After writing, we deliberately bracketed them to prevent interference and improve the ability to contact the primordial concreteness of lived reality. 

### 2.2. Gathering of Experiential Material

The quality of the collected experiential material is essential for fuller access to the intended lived experience. In light of this, participant recruitment and selection occurred in a Portuguese psychiatric department of a general hospital in a metropolitan city. Ethical procedures involved consecutive steps as follows:Obtaining authorization from the clinical director and head nurse of the psychiatric department;Obtaining approval from the Hospital’s Board of Directors and the Ethics Committee;Guaranteeing participants’ informed written consent assuring their anonymity and confidentiality;Finding a place that guaranteed discretion and a favorable environment for experiential sharing;Transcription of the experiential material from the audio-recorded interviews replacing or deleting all names and locations;Storage the verbatim and audio files so that it was only accessible to the researchers.

Another fundamental aspect pertains to the inclusion criteria, which consisted of participants between 18 and 64 years of age with only one admission in a psychiatric service, who, after hospitalization: Returned home;Resumed activities compatible with an active life (people who carry out work or school/training activities, including volunteer work);Spoke Portuguese fluently;Had outpatient follow-up by the psychiatric department;Agreed to participate in the study and signed informed consent.

In order to prepare the recruitment location, a presentation to the multidisciplinary team of the psychiatry service became essential. Afterward, participant recruitment involved collaboration from the medical team in selecting possible candidates based on the outpatient consultation of the department. As referencing emerged, we contacted participants via mobile phone to make a brief presentation of the study and propose participation. Quickly we realized that people demonstrated reduced adherence in coming to the outpatient service to carry out “just” an interview. The first three contacts refused to participate in the study, entailing a need to change strategy. Hence, we combined the interview with a psychiatric or nursing appointment, which implied greater adherence and facilitated access to the participants.

We conducted fifteen phenomenological interviews with twelve participants (three expressed availability for a second interview, allowing for a greater exploratory scope of the lived experience). The total number of possible participants contacted comprised twenty-three people. The reasons for rejecting participation were: unavailability of time, no-show on the scheduled date for the interview, previously accepted participation but rejecting it at the time of carrying it out, or personal unavailability to recount the events associated with this lived experience, characterized as too painful. 

The interviews occurred in an outpatient office of the psychiatric department, lasted from 27 min to an hour and 21 min, and we used an audio recorder to guarantee the truthfulness of these testimonies. The interviews started with the initial request-to speak about the experience of returning to daily life after psychiatry admission. A nuance of conversation and informality characterized these unstructured interviews. However, as they went on, it became conducive to requesting a specific experience, focusing on what, when, and how, and soliciting concrete details about what happened, the lived thoughts, and feelings at the moment. After the interview, we wrote field notes to record the what and how of the shared narratives, disclosing the context in which it occurred, the shared episodes, and the ingrained impressions and reflections. We collected experiential material for six months, specifically from 5 June 2018 to 18 December 2018. The verbatim transcription took place concomitantly and subsequently to this entire phase.

A total of four women and eight men, ages between 22 and 46 years old, participated. People with psychotic disorders were the majority (*n* = 9), with the remaining three presenting a mood disorder, which also had psychotic symptoms at the time of admission. The timing between discharge and the interview goes from two months to seven years. Regarding education, three women and three men had university-level education, and one female and five male participants had secondary-level education. For marital status, one man was married and had two children, one woman was divorced and had one child, and the remaining participants were single, although one woman was living with her significant other, and two men had a romantic relationship. Finally, three women and five men were employed, one woman and two men were students, and one man did volunteer work.

### 2.3. Thematic and Hermeneutic Analysis

After transcribing the interviews *verbatim*, a careful process of familiarity with the material began through repeated listening and reading the interviews to obtain a global sense of the phenomenon. The four epoché-reduction methods proposed by van Manen (heuristic, hermeneutic, experiential, and methodological) conducted this primary holistic approach [26]. An intertwined gesture of wonder, openness, and concreteness supported the exercise of flexible rationality, methodologically informed, regarding the essential notions and foundations of phenomenology, whose *raison d’être* is to illuminate aspects of pre-reflective life. 

After this familiarization, it became necessary to activate the mechanisms of the reduction-proper driven by an eidetic, ontological, ethical, radical, and originary reduction. The interviews began to be edited, which, initially, consisted of a selective reading approach highlighting significant value sentences, removing accessory material with no vocative value, such as narrative redundancies and “paraphenomenon” descriptions, and organizing temporally and situationally the presentation of the different experiential episodes described by each participant. That allowed for greater conciseness about the empirical reports of lived experience that resulted from the phenomenological interviews. After that, the edition entailed elaborating on the phenomenological anecdotal examples according to the indications provided by van Manen [26]. Each anecdote consists of a succinct narrative style, usually describing a single situation, where experiential concreteness is emphasized (with quotes about what was said, done, and felt). They must start close to the core of the experience for a rapid climax of apotheotic clarification, i.e., a *punctum* [26]. 

The writing of these anecdotes resulted from a detailed reading approach where line-by-line, word-by-word, we reflected on the embedded meanings and sough the experiential concreteness presented in the captured phenomenon. Supporting this entire process, the QDA Miner Lite^®^ v.2.0.6 software facilitated the organization, editing, and, above all, an approach to narratives in a holistic, selective, and detailed way, whose eidetic apprehension substantiates the thematization and writing of the phenomenological text, concerning the experiential material involved. This thematic and hermeneutic analysis and reflection was laborious and had the irreplaceable contribution of a collaborative group made up of five experts who kindly welcomed and provided the perspectives of all elements in a space of mutual and constant exchange of knowledge and thoughts about the phenomenon but also the method used to capture it. That brought an innovative and luminous perspective to analyzing the experiential material, adding a sensitivity of profound interpretation of this lived world from these other perspectives, increasing the scope of the “phenomenological lenses” of this study. In addition, insight cultivators became powerful sources for thematic insights and phenomenological writing. These consisted primarily of seminal books on phenomenology and literary works, including authors such as Martin Heidegger, Maurice Merleau-Ponty, Karl Jaspers, Jean-Paul Sartre, Paul Ricoeur, Milan Kundera, Virgílio Ferreira, and Fernando Pessoa.

### 2.4. Writing of the Phenomenological Text

So, a thematic collection started to emerge and shape this phenomenon’s gestalt with a back-and-forth movement of writing and rewriting, supported by an iterative analysis and reflection between the parts and the whole. We wrote the phenomenological text as the thematic and hermeneutic analyses were conducted. The separation between these investigative movements proved impossible since the material’s analysis and reflection appealed to the intent of writing and rewriting that continually recreated and sought the constitution of original structures of meanings. Moreover, it was critical to achieving a textual vocative dimension since human experience goes beyond the tangible shapes it defines. So, writing the phenomenological text engaged the vocative method that gives texture to the noncognitive and pathic knowing of the lifeworld [26]. 

Hence, the vocative method of phenomenological writing entails: A rational modulation as iteratively and systematically explores the meaning structures of the phenomenon;A non-rational modulation because it recognizes the need to find or produce expression mediators that provide access to the antepredicative experiential nature of the phenomenon [26].

Hence, the philological methods inspire the researcher to present multilayered prose that intends to mimic the phenomenon’s experience through a resonance that leads to the recognition of the experiential plausibility of that phenomenon, even if the reader has never experienced it [26]. From the inweaving of the five philological methods proposed—revocative, evocative, invocative, convocative, and provocative—great care and thoughtfulness went into rooting the findings in first-account anecdotal examples that illustrate the thematic structures that guide the essential facets of the phenomena under study.

## 3. Results

In the capture of the original essences of the researched phenomenon, seven themes emerged from the participants’ narratives. These shape up the dynamic and interpenetrating conjugation that reveals the incarnated constitution of these participants returning to daily life after the first psychiatric hospitalization. Moreover, these themes revolve around two constitutive poles-the self and its relationships-as four themes are person-centered, and three themes illustrate the relations that constitute the “thrownness” entanglement of Being-in-the-world. Therefore, the first four themes regarding the experiences of returning everyday life respecting the self are: (Un)veiling the imprint within the self;The haunting memories within the self;From disconnection to the assimilation of the medicated body in the self;From recognition to overcoming the fragility within the self.Towards the relational pole we have:The relationship with health professionals: from expectation to response;The relationship with others: reformulating the bonds of alterity;The relationship with the world: reconnecting as a sense of self.

As no hierarchical order is inherent to the themes, we present primarily the themes related to the self, followed by the remaining three. In order to establish a clear relation between the *datum* and the findings, illustrative quotes from the participants’ phenomenological anecdotal examples are summarized in two tables, respectively pertaining to the already mentioned constitutive poles-the self and its relationships. 

### 3.1. (Un)veiling the Imprint within the Self

This theme focuses on the experiences these participants underwent regarding the revelation they decided to make about the experience of mental illness and ensuing hospitalization. The (un)veiling captures the extent to which participants make the revelation that goes from wholly veiled to the unavoidable unveiling. Another striking word of this theme is—imprint—whose meaning does not fit here the tangibility that a mark can have on a person, but rather the impression left by something (more or less indelible) that substantiates the embodiment of Being-in-the-world. 

Several circumstances underlie the disclosure and allow participants not to mention anything about their mental illness and hospitalization, such as the fact that people did not notice that they were in the hospital for several days or did not witness behavioral changes of the mental illness that took them to the hospital. In this sense, some participants did not have to justify their absence and decided not to share anything related to the psychiatric hospitalization or their mental illness, namely at work, school, and with relatives. Otherwise, participants reveal that to justify their absence (from daily living), careful consideration went into the extent to which they disclosed their psychiatric hospitalization. Therefore, unveiling comprises several layers of revelation as participants speak of a “nervous breakdown”, heart problem or stroke instead of a “psychotic break” or hospitalization due to a mental illness. Indeed, such a layered approach arises from the stigma and discrimination still attached to a psychiatric inpatient unit that convokes the self’s need for protection. The associated meanings of psychiatric hospitalization, such as shame due to the responsibility felt towards admission, and the label associated with people that need psychiatric hospitalization, linger and affect its disclosure. As a result, most participants present a selective and careful approach as they ponder whether or not this revelation makes sense. 

For other participants dealing with the imprints of mental illness emerged as unavoidable as the illness entailments resulted in disturbances in everyday life before hospital admission—due to symptomatic manifestations of mental illness-or after hospital admission—due to iatrogenic effects. Several forms of coping with these repercussions come through as participants try to keep their integrity and avoid this unveiling, masking the signs that could give away their mental illness *pathos*, namely when resuming previous activities. Nevertheless, the experience of stigma and discrimination took place with some participants that had to deal with social discredit and its repercussions.

Even though stigma and discrimination still exude from the disclosure of mental illness, such as schizophrenia, or psychiatric hospitalization, for some participants the need to express their authentic self to others overpowered such risk, namely when participants engage in a romantic relationship or consider their disclosure may help others who are also experiencing mental suffering. Quotes for this theme are provided in Table 1.

### 3.2. The Haunting Memories within the Self

From the experience of mental illness and psychiatric hospitalization, memories inadvertently invade the lived present, configuring it in such a way as to attribute a form whose meanings influence the being to be. Firstly, psychiatric hospitalization emerges as a traumatic event which calls for the recognition that hospitalization is an event that can have significant repercussions on the individual such as: no longer tolerating the image of hospitalization, fearing the return to the space of psychiatric admission for a psychiatric consultation, inhibiting the attendance of other health services for having to return to the hospital space, or being a nightmare when it invades dreams.

The space where mental illness manifested itself also emerges as uncomfortable since the resulting memory brings the person closer to this maladjusted past by the mental illness, and this confrontation triggers a reliving of the situation that the space evokes. This revival can even appear in objects associated with the internment, such as tea, whose memory transports the person to the perceived body, impaired by medication side effects, which did not work even in the most elementary way (such as chewing).

Time and its peculiar seasons can also bring to light meanings triggered by memories, such as being hospitalized during Christmas, a festive and happy time, which also incorporates a participant’s memories of the hospitalization that happened the year before.

Ultimately, memories related to the psychotic break, as an existential possibility of the self, render a terrifying shadow, which limits the possibilities of the present self. Quotes for this theme are provided in Table 1.

### 3.3. From Disconnection to the Assimilation of the Medicated Body in the Self

The experiences reported here show the perception of a disconnection that accounts for the mismatch between the habitual body (pre-hospitalization) and the actual body (medicated, post-hospitalization). Indeed, Merleau-Ponty’s “own body” concept arises throughout this theme as it intertwines the general with the contingent experience of the world. In other words, inhabiting the own body, we find “the habitual body-that of general and pre-reflexive existence-from the actual-that of personal and reflexive existence-understanding that both always co-penetrate each other” [29] (p. 2). Despite the felt liberation with the hospital discharge, participants face their own body that is out of step with the possibility of full existential realization. 

Even though some participants recognized relevant therapeutic effects, the medication imposes profound changes that combine feelings of unfamiliarity, non-identification, and non-realization of themselves in the world. 

This disconnection implies a lengthy and challenging process of therapeutic adjustments that impairs the person’s ability to fit in the world, requiring a successive updating of the medicated body in the own body.

Considering the disconnection experienced in the participants’ own bodies regarding the conciliation between the habitual body (pre-hospitalization) and the actual body (medicated, post-hospitalization), reports of resignation towards the need for medication emerge amid doubts.

Nevertheless, one of the participants stopped taking medication. However, after psychotic recidivism, medication gained another perspective to him. The fear of having another psychotic break and risking his job, relationships, and sense of control, brought to medication a renewed sense of purpose.

Additionally, it became clear that participants may consider that there is a limit to what their own body can accept regarding the iatrogenic effects of medication and its nefarious entailments in daily life. Therefore, it became essential to continuously reconcile the medication to the patient’s perceived needs to guarantee that mental illness did not manifest itself again and that patients could retain a sense of accomplishment and fruition. Quotes for this theme are provided in Table 1.

### 3.4. From Recognition to Overcoming the Fragility within the Self 

In this theme, the component of the fragility of human existence appears taking on a specific form—the possibility of mental illness leading to readmission that participants actively strive to overcome. A sense of control over a pervasively persistent doubt begins when recognizing the fragility within. Each of these participants defines ways of overcoming it, understanding the contingencies that can put them at the mercy of this fragility. Hence, the definition of strategies and certain precautions allows them to gain awareness and curb its manifestation as they now recognize the fragility that exists in themselves.

Inherently, a belief emerges, a conviction of being able to tame this fragility through the knowledge of their limits and potentialities, which leads them to become more aware of the repercussions that certain events or actions can translate into themselves.

Ultimately, participants describe this process as a catalyst to foster personal growth toward a perceived higher maturity of the self. Quotes for this theme are provided in Table 1.

### 3.5. The Relationship with Health Professionals: From Expectation to Response 

The relationship with health professionals stands out in the participants’ narratives with a double co-responsiveness established between the person and the health professional, but also between the expectations and responses that both actors reciprocally attribute to each other. A magnanimous purpose guides these relationships: to restore the maximum level of health that the person in need of health care can obtain. From this intentionality, the health professional inhabits the world of these people who return to their daily lives. Furthermore, admission marks the beginning of consistently using mental health services and consuming psychiatric medication for all participants.

Establishing a relationship with the psychiatrist appears inevitable, entailing an explicit aspect of commitment regarding medication, in which several participants express their side effects as upsetting and even question its need. The psychiatrist assumes an attribution to judge the person’s evolution, deciding the stipulated medication plan. To support his credibility emerges the belief that he is the holder of what the participants consider to be higher knowledge about mental illness and its treatment.

The relationship with the psychiatrist can be considered enough regarding the participants’ follow-up to a certain extent. Four participants expressed the support given by the psychiatrist as sufficient. However, several participants felt it was insufficient, and other health professionals emerged to also intervene in the therapeutic project. The reasons for their involvement entail the need to have someone with whom to vent, with whom to talk about the experience of mental illness without shame, and with whom to share their fears, concerns, doubts, and uncertainties. These aspects stand out in these participants and call for a multidisciplinary team that can ensure the response to what are also relevant expectations on the part of these participants.

In all these relationships, trust, appreciation, permanency, and relational co-construction are fundamental for establishing a relationship that favors the symbiosis between expectation and response. Moreover, health professionals’ duty of neutrality seems to appear as an expectation on behalf of participants. Not only towards the preconceive ideas participants feel society, in general, may have, but also towards family members. Quotes for this theme are provided in Table 2.

### 3.6. The Relationship with Others: Reformulating the Bonds of Alterity

From the unfolding of the narratives that shape this thematic essence, there is a horizon of reformulation of the relational quality established with the various others in their daily lives, entailing greater intensity in those that we could call significant relationships.

Particularizing the reformulations concerning parents and close relatives (like siblings), this horizon comprises ways of reformulating this otherness, understood as supporting, securing, reinforcing, and bringing together previously established bonds. Hospitalization brings forth the opportunity to reestablish relationships as the unveiling of mental illness and consequent hospitalization set in motion a perceived fragility that reshapes these participants’ alterity.

Nevertheless, for other participants this parental relationship appears as intrusive, reducing the sense of autonomy and self-efficacy perceived by the person, creating an asymmetry in the relational transaction.

Some participants acquired a sense of responsibility for the repercussions of their illness on their parents and actively sought to live up to what they understood to be the expectations they had for themselves or to remedy the suffering they believed had caused.

It is also possible to apprehend a participant’s sense of maturation towards the parents, entailing a reformulation of the intersubjectivity, which comprises the transition from child to adult.

Another significant relationship relates to the spouse, as mental illness’ repercussions affect the ability to correspond to the roles and expectations defined upon returning to daily life. For this reason, the relationship plunged to a breaking point, resulting in a reformulation of a love relationship into a friendship relationship.

Regarding previous relationships, participants express satisfaction for perceiving that they are again recognizable by others as themselves after facing a kind of non-sameness that prevented them from relating to people. Conversely, some relationships could no longer ensure their maintenance or maintain the same relational quality. Respectively, in this reformulation of alterity emerges the end of damaging relationships, namely friendship relationships with schoolmates or even friends, with participants establishing completely new relationships.

Finally, it is necessary to designate the relationships that took shape in the post-hospitalization period precisely with other people hospitalized simultaneously and who constituted themselves as bridges for expanding participants’ reach to others. Quotes for this theme are provided in Table 2.

### 3.7. The Relationship with the World: Reconnecting as a Sense of Self

This last thematic essence carves the human being’s root in the world in which his thrownness acquires protagonism. Participants elucidate the co-construction that shapes every human being towards fulfilling the desire for a sense of self in the world. “The world belongs to Being-one’s-Self as Being-in-the-world.” [28] (p. 146) Despite the apprehension that may have arisen after discharge, participants yearn to reconnect to constitute once again the essence of being that exists and, therefore, engages the world.

Some participants experienced returning to daily life as a void, a world without direction, impoverished for not having spaces to go to other than their home or health services. Accordingly, feelings of loneliness and boredom stand out as medications’ side effects also disturb participants’ ability to leave out into the world again. Notwithstanding, facing existential emptiness, there is an ambition for activity, structure, and even routine that gives a purpose and sequence to the days and allows us to enjoy meaning in our acts. There is a clear call from the world that projects participants out of their empty and disconnected existence. Gradually, even in trembling steps, the will to gain a sense of fulfillment acquires impetus and launches them towards “the authenticity of Being-one’s-Self” [28] (p. 229). Quotes for this theme are provided in Table 2.

## 4. Discussion

The results of this study indicate that the return to daily life after the first psychiatric hospitalization constitutes a transitional challenge that places the person in front of adversities that she gradually manages to overcome. So, things are not immediate, and there are advances and setbacks in this transition, which is complex, idiosyncratic, and challenging. Accordingly, the literature has pointed out this finding reinforcing that discharge is not a clinical point but a process including more than the formal moment of leaving the hospital with the implied follow-up recommendations [2,10,11,13,18]. As participants give voice to their struggle, commonalities in this particular lived experience have a significant expression in all the themes.

The theme “(un)veiling the imprint within the self” includes a strong influence from stigma and fear of social discrimination exerted on the person who determines this (un)veiling, taking into account:Their own idiosyncratic needs to fully share their history with the other;The relational nature established with the other (whether it is a romantic, professional, or friendship relationship);The situational context that calls for this sharing;The inevitability felt towards the other, who calls for clarification of what happened;or, The revelation naturally emerged from the dialogue established with the other.

Nevertheless, an underlying perceived stigma, particularly psychiatric inpatient admission, lingers and shapes the disclosure extent. Several studies have stated that psychiatric hospitalization has an inherent social stigma, which seems to put a considerable toll on people’s identities [2,9,13]. So, being in a psychiatric hospital or unit carries out a statement about the seriousness of mental illness that affects how patients decide to share their identity with others. Concurrently, when the person is aware that she had inappropriate social behaviors resulting from mental illness, or iatrogenic effects that affect the ability to interact socially, this stigma gains even more projection.

Consequently, the management of pre-assumed expectations regarding disclosing hospitalization to other people becomes evident, as already described in the literature [9,10,13,18]. Thus, from the decision to fully disclose what they experienced (simultaneously accepting and fully assuming their history) to total concealment or omission of their admission, in order to not disturb or damage relationships, sets the range.

What is noteworthy here is the description of discriminatory events that the participants share after revealing their admission to the psychiatric inpatient unit. This situation arises in the current literature [9,13,30], which should concern society in general and each of us as human beings, but also healthcare professionals. The maintenance of negative responses regarding sharing the experience of psychiatric hospitalization should continue to be the target of a global action with all citizens, but also with patients who may need support in establishing strategies that prepare them to challenge social stigma related to disclosure, as well as to manage the ensuing repercussions [9]. It also seems essential to invest in empowering people with mental illness, reinforcing their beliefs regarding their abilities and mastery over their life. Hence, this means fostering self-esteem, perception of self-efficacy, and problem-solving ability, clarifying that they have the right to establish a relationship in which they feel accepted as a person of value and feel free to express themselves without being held back by fear of being rejected [31,32]. The almost idealistic ambition of these goals highlighted here is recognized. Perhaps they do not even constitute feasible specific objectives such as a goal in which, after being surpassed, the effect of its conquest forever remains. Many people—even those without mental illness—will (or have) find it challenging to recognize and accept their value. Like many other people, it is difficult to express yourself freely without fear of being rejected. For this, mental illness is not an essential condition, but it worsens the possibility of developing well-being and more-being. More than an objective, it should be an inspiration that helps people to walk more and more towards who they really are and who they can still be.

Another central aspect of this phenomenon is that of the own body, whose challenge shines through the theme “From disconnection to the assimilation of the medicated body in the self.” In this, the disconnection experienced by the medicated body installed in the present-actual body- is remarkable. This disconnect essentially comes from medications’ iatrogenic effects. Here, the need to understand that the expected results may not be quickly attainable counterbalances with the hope of an attainable recovery.

Like in other studies, these participants describe the sedative, slowing, thought-emptying, libido-reducing, depressive, dragging, heavier, and even painful effects they experienced with taking the medication [10,11,12,13]. It is relevant to note that all these participants showed psychotic symptoms at their hospital admission, and some were able to express insight. Therefore, despite iatrogenesis, it is also possible to listen to their awareness of the added value implicit in taking the medication—not having psychosis. However, some participants established a limit for their iatrogenic tolerance risking medication adherence. They accept suppressing their possible psychotic “condition” as long as this does not imply suppressing what is essential for their lives. This specific finding is paramount as the literature (to the best of our knowledge) had not yet clarified the embedded meanings inherent to the lived experience of people who are faced with such a predicament, even though motivations for dropping the medication have been listed such as feeling it is no longer needed, it does not work or causes exceeding side effects [12,13,33].

The debate on medication in psychiatry is central since it plays a vital role in the person’s good prognosis-despite not constituting an exclusive therapeutic option-and involves a balance of pros and cons that are fundamental to define the limits between symptoms of mental illness, the effects of the medication, and one’s personality [34]. To what extent will patients be willing to sacrifice what they consider essentiality of their personhood to reduce the risk of a possible new psychotic break, particularly when they are not psychotic, when they have their full mental faculties? Therefore, factors like poor satisfaction with the treatment provided, less medication supervision at home, negative attitudes toward medications, or poor quality of the therapeutic alliance constitute critical elements to medication adherence, implying the need for their assessment during outpatient treatment [35]. Hence, the assimilation of the medicated body into the actual body is fundamental for realizing a successful transition process that is inevitably inherent to the return to daily life after the first psychiatric hospitalization. Otherwise, patients will deal with immense suffering and restriction in fulfilling their lives. The need for a transversal and specific intervention for people experiencing this phenomenon has to integrate provided mental health care to guarantee continuity of care that acknowledges the inherent transitional challenges [7,10,18]. Accordingly, the literature points out the development of several interventions addressing this issue which, broadly, fall into the denomination of discharge or transitional interventions. All bear in mind the intrinsic vulnerability of psychiatric discharge and transition to the community [36,37]. However, variability of interventions, outcomes, and outcome measurements produces a low level of evidence, undermining recommendation [37]. Nonetheless, transitional interventions with bridging components constitute a preferred care delivery structure by service users, i.e., interventions which embrace the movement from inpatient to outpatient service, including pre- and post-discharge interventions [37,38]. That being so, transitional interventions require further research addressing homogeneity of outcome reporting but also deserve recognition as alternatives or complements to usual or no discharge plans [36,37].

In addition to participants’ own body, a place where everything is fulfilled [39], memories, composing a stronghold of the experiential recollections of life’s fulfillment, stands out in the primordial sensation of the lifeworld of these participants.

The theme “The haunting memories within the self” exposes how the deposited memories of the internment stir the lived present and affect its future. The present experience is better understood when reflected in the mirror of history. What history conveys to us comes alive in light of our time which conveys a reciprocal clarification of the past and the present [40], i.e., there is still some clarification to be gained for these participants. What help do we want to give the person in this clarification? Here is personified the immense intensity that admission in psychiatry imprints in memories and delegates in the path taken in the future. It is an appeal that deserves, even demands, a response that, for the time being, is not recognized in mental health care [7,10].

Particularly concerning nursing care, if we consider, as an example, community home support services, stipulated for nurses in Portugal, involve the promotion of autonomy in basic and instrumental life activities, access to occupational or recreational activities, awareness and involvement of informal caregivers, and medication supervision and management [41]. Overlooked are the repercussions of the lived experience that affect and determine patients’ potentiality for achieving enjoyment in life. So, interventions focused on the emotional processing of disturbing experiences (even described as traumatic in this study) are needed [1,14,15,42]. The implementation of trauma-informed care deems relevant as reports of trauma exposure in inpatient psychiatric services emerge in the scientific literature [14,15,42,43].

Regarding interventions, “The relationship with health professionals: from expectation to response” shines through in every participant’s narrative.

The psychiatrist emerges emphatically associated with the treatment’s definition of the what, how, when, how much, and why. In what is evident as a cunning and complex dance between expectations and mutual responses between the dyad doctor-patient, variability characterizes participants’ lived experience. Although perceptions of support, encouragement, sharing, even to an almost friendship stand out, there is also disappointment when it is from the doctor that the person feels compelled to do something that she does not want or fully understands—medication. Other studies reveal this attrition between what the patient considers a need for medication adjustment considering the side effects experienced and its impairment on daily living [7,12,13,33]. Hence, the management between expectation and response necessarily implies a constant adjustment of reciprocal conciliation between this dyad that finds, in each of its elements, the availability to make concessions and define solutions that can give in return—response—the promise of a good life but without compromising the person’s health and life expectations.

It is fundamental to highlight how some participants expressed the need to have a follow-up from someone, such as a nurse or a psychologist, with whom they could talk, vent, and get support to acquire meaning. The non-pharmacological component of the treatment is essential in the response expected by these participants. The need to talk to someone, share feelings, and recognize and be recognized by another from a considered neutral setting, operates as a fundamental therapeutic component in the transition process involved in this phenomenon. The urgency of a helping relationship in this return to daily life finds an echo in the various health professionals involved here—nurse, psychiatrist, psychologist—which makes sense to ensure in this specific transition that takes place at different levels: from psychiatric hospitalization to daily life; from the mental illness crisis to the remission of the symptoms expressed by the mental illness in the person; from an inpatient to an outpatient setting. The literature has made this necessity clear [2,10,13,17,18] and constitutes a real alert for its transversal integration in the transitional care stipulated by the health services.

The nurse stands as a resource already recognized as competent and necessary to make more available to the person with mental illness [2,12,43]. The versatility that characterizes her training and performance within multidisciplinary teams becomes an asset in this context of clinical practice. Nurses’ interventions allow for a systematic approach to different problematic focuses manifested by the person, from the insufficiency or inability to satisfy their fundamental human needs to overcome the difficulties arising from situations of health-illness that invoke the person’s growth in terms of their ability to deal with increasingly complex situations and thus acquire their full human potential. This integrality and contextuality of the nurse’s action, which is inherent to humanist nursing care, gives insight into the uniqueness that the nurse-client encounter implies [25]. In a nursing conception of caring–healing, the return to daily life after hospitalization in psychiatry fully embraces this healing component through time, difficulties, and idiosyncrasies involved in the person’s recovery.

Moreover, the contribution that the mental health specialist nurse can introduce in this situation resides in the developed specific skills in establishing therapeutic relationships, mobilizing psychotherapeutic, socio-therapeutic, psychosocial, and psychoeducational interventions [44]. Understanding the dynamism and synergies between the parts and the whole, nursing’s action of caring-healing comprises a clear systemic and holistic vision that mobilizes and integrates a diversity of resources that optimize the establishment of well-being, but also aspire to a subsequent level of more-being [25].

In need to generate meaningful and significant relationships, from which health professionals also find their share, relationships with others in daily contexts get through. “The relationship with others: reformulating the bonds of alterity” is a theme that gives an account of the primordial humanizing power that the other has in the life of the self. The presence of meaningful and fruition-giving relationships is highly relevant for these participants, who clearly express this need for the other. The literature consistently expresses this need that increases feelings of security, well-being, and hope [2,9,13,16,18]. At the same time, the perceived absence of this support can constitute a problem, which often leads to the adoption of ineffective coping strategies, contributing to later readmission [9,10,13,16,18]. In this theme, it is worth highlighting the immanent reformulation in which the person starts to manifest or feel the alterity before the face of the other, often a family member. In the experiential thematic diversity, family constituted a source of shelter and security, supervision and caution, and a source of growth and reparation of broken family ties.

Nevertheless, the quality of marital relations emerges as problematic since mental illness’ toll can affect their maintenance. As side effects hinder compliance with expected roles and tasks, family life becomes more burdensome, mainly for a spouse and/or a parent [7,10,11,13]. Self-efficacy garners a central role in developing the coping skills necessary to adjust to daily life’s responsibilities and demands [10]. Therefore, assessing and intervening in the scope of self-efficacy may be relevant to this transitional phenomenon. Likewise, including relevant relationships, such as familiar ones, in discharge planning promotes awareness, inclusiveness, and hope among relatives [2,16]. Unfortunately, discharge from inpatient psychiatric units still appears to insufficiently integrate patients’ voices and coordinate communication between health and social care professionals [45,46,47].

Other relations such as friends, colleagues, and even relationships established and later maintained after hospitalization constitute a source of emotional support [10,12,18]. More importantly, the literature refers to support groups as helpful mainly by the opportunity to vent with people that shared similar experiences [10]. Accordingly, peer support facilitates access to local communities, and promotes friendship, providing a space for understanding and encouragement [2,10,37]. As a result, this relational reformulation, that these participants experience, is most distinctive in this theme, leading to closer and more satisfying relationships or to relationships with greater tension and dissatisfaction. This possibility should be a focus of attention by health professionals, namely nurses, as the relationship with the other constitutes a fundamental human need, a motto for a fuller reach of the human potential whose fulfillment goes beyond oneself demanding the authentic encounter with the other.

Complementary to moving towards the other, “The relationship with the world: reconnecting as a sense of self” completes participants’ need for exteriority beyond their home. This achievement is particularly arduous for those who, after discharge, cannot return to the activities that previously filled their day or have difficulty establishing/complying with meaningful activities due to iatrogenic effects. The recovery process’s coexistence with medication’s side effects translates into a readjustment to everyday life which entails the need to engage in meaningful activities, which include: structuring of the day’s schedule (a routine), an assumption of responsibilities, a useful or pleasant occupation of time, an opportunity to regain a certain sense of normalization and integration in society [10,11,12,13,18].

The theme “From recognition to overcoming the fragility within the self” gives an account of the meanings attributed to the fragility experienced that fills with uncertainty, various moments of the return to everyday life, and longing for recognition and desire to overcome. This theme resonates with the scientific literature that describes the perception of people living with a lurking mental illness [13,17]. The possibility of relapse remains as a “ghost” closely linked to the haunting memories that permeate this return to these participants’ daily lives. However, there is a declaration of a belief and hope of overcoming this fragility through coping strategies such as:Greater self-surveillance;The need to keep a small dose of medication;Avoiding triggering or precipitating factors associated with psychotic thoughts;Identifying the prodromal signs of a crisis;Focus on reality and the sound and fundamental things in life;Looking for a healthier lifestyle;Keep the mind focused and concentrated on studies;Recognize when feeling worse and ask for help.

All these coping strategies culminate the efforts that these participants made to deal with the fragility within themselves and achieve a return to not only home but to everyday life. These are actual examples of overcoming that may provide clues for structuring transitional interventions that could constitute a resource for others in a similar situation. Indeed, “without being able to experience coping, some service users are unconvinced that they can cope unaided in the long-term and that interventions are palliative rather than restorative” [12] (p. 366). Consequently, knowing what to expect and sensing control towards life, through the development of coping strategies, allows for a more fluid and healthier transition [47,48].

### 4.1. Limitations

Although this study has always sought to remain faithful to the methods proposed by the phenomenology of practice, its results must be considered cautiously, and their generality is not the aim of the investigation that we have concluded here. Likewise, it is noteworthy that there were people who refused to participate in the study precisely because they considered it too painful to recount what was the experiential constitution of returning to everyday life after the first psychiatric hospitalization. In this way, there is a limited experiential breadth due to not having access to the reports of people who expressed fear of unacceptable suffering in recounting their lived experience.

Another caveat regarding the study’s limitations concerns the answer to the research question that triggered this entire work. This answer is not final or watertight, something always inherent to a phenomenological study, but it is also not comprehensive enough for the intended objective. These participants have different characteristics, but one is transversally present: they all experienced psychotic symptoms, and the diagnostic diversity is short. The literature has exposed that variables such as age, sex, medical diagnosis, and stage of recovery can change the lived experience and the manifestation of needs [10,13,49,50]. Therefore, we consider it essential to emphasize that there may be people whose specific characteristics, namely the medical diagnosis, can change the lived experience and its experiential meanings. Nevertheless, the orientation of this work is phenomenological, and as such, there are no absolutes. There is only the whole particular infinite that each person is and builds in herself.

### 4.2. Implications for Practice and Research in Nursing

The peculiarity of this study consists of focusing on people who had a single hospitalization in the psychiatric service and managed to reconnect with the world, having active participation in the surrounding contexts beyond the limits of their home. Undoubtedly, the phenomenological findings presented here can constitute a basis for establishing transitional care, which must be transversally ensured. This care must go beyond symptom control and have a concrete translation in the support provided by health professionals to the expectations or needs identified by patients (and not only by professionals).

Nurses, particularly specialists in mental health and psychiatric nursing, are present in the health care contexts through which these individuals pass, often in close proximity, and have the duty, but also the competence, to strive to acquire the conditions and resources needed to develop humanistic and integrative nursing care, centered on the person and the recovery process. These can be constituted as the facilitators of a transitional care model that guarantees a dynamic interaction between hospitalization and community services, assuming the scope that treatment must include and that it is not limited to symptomatic crisis and remission but also encompasses the healing and recovery of patients [50]. Specifically, the meaning and purpose in life despite mental illness and its treatment consequences [51].

It is essential to reflect on this since local mental health services are the basis of the Portuguese mental health system, and its organization is built precisely in the form of a general hospital’s department or service, according to the respective area of influence [52]. Therefore, the hospital constitutes the core of mental health care, where care provision is defined and ensured for a given population instead of a community infrastructure.

We believe that this work can facilitate the awareness that it is necessary to operationalize the paradigm shift in the contexts of clinical practice and that there is a human potential already installed in the field, which can provide greater robustness and response to the needs of the person who returns to his/her life daily life after hospitalization in psychiatry [53]. Specialist nurses are certainly an asset and have the necessary skills to prepare and ensure many of the necessary responses. These can play a crucial role in this desideratum, as they are aware that regardless of the origin of the transition, its progress can be mediated by the support of the provided environment [48]. Establishing therapeutic relationships in a collaborative partnership, considering the individual’s choices, experiences, and circumstances is a valuable contribution that mental health nurses can provide [54].

Regarding the implications for research, we believe there is still a need to invest in qualitative research (phenomenological) that will bring clarification and sensitive knowledge to clinical practices’ action. Within the scope of the phenomenon studied, we consider that there is a need to describe the lived experiences of people with different sociodemographic profiles, namely medical diagnosis, length of stay, or even type of hospitalization—whether voluntary or involuntary. Of course, the investment in the development and evaluation of the effectiveness of interventions specific to this transition becomes evident, assuring the inclusion of a person-centered approach so that an appropriate pace favors a positive view of the discharge process and a continuum of care [55].

## 5. Conclusions

The narratives of the lived experience about the return to everyday life after the first psychiatric hospitalization elucidate how individuals deal with profound changes in their lives. Therefore, it constitutes a transitional, complex, and demanding challenge to which health professionals must be attentive and prepared to act in complete esteem for the needs, uniqueness, and willpower expressed by the person. Stigma and discrimination still act on people’s lives and require the empowerment of individuals and their families and the management of pre-assumed expectations regarding disclosing hospitalization or mental illness to others. Mental health services still have to reinforce the community model by considering placing outpatients’ appointments in different locations than inpatient units (when positioned in the same place). Additionally, transversal implementation of transitional interventions with bridging components between inpatient to outpatient units appears essential. These should gradually reduce the support given by health professionals, consider integrating relevant relationships, such as familiar ones, trauma-informed care, peer support, and the development of adjusted coping strategies to foster self-esteem, perception of self-efficacy, and problem-solving ability. Health professionals, namely mental health specialist nurses, can have a pivotal role in establishing a helping relationship with recovery-oriented goals, coordinating patients’ transitional care, and assuring continuity of care sensitive to the subjective experiences, volitions, and resources expressed by the person.

## Figures and Tables

**Table 1 jpm-12-01938-t001:** Illustrative quotes for themes regarding the constitutive pole of the self.

Themes	Illustrative Quotes
(Un)veiling the imprint within the self	“When I talk about this subject, it’s with people I see who try to understand. It is not with people chosen randomly. They are handpicked.” (P1) “When it makes sense, I speak up. It’s not a secret I hide. It’s also not something I’m going to say in the public square because I know there’s a lot of stigma.” (P2) “Some colleagues from work know I had a nervous breakdown. They don’t know about the psychotic break. I say that to some people, not to all.” (P2) “So as not to harm my son’s thoughts or my career, it was said that I had a health problem, something with the heart, I was hospitalized.” (P4) “I never shared any of this psychiatry stuff at work to avoid any conflict of interest.” (P5) “I decided not to tell anyone and to be a thing of the past.” (P6) “Maybe I’m a little ashamed to say that I was hospitalized. It’s because I was smoking (weed) badly, alone, and stupidly, and that was my mistake. I feel angry at letting myself be fooled by my own stupidity. There is no justification. And the other part has to do with the labels given to people who have some kind of psychiatric problem and what I feel that other people eventually feel about me having been hospitalized in some psychiatric hospital unit rather than being in a clinic. The psychiatric hospital is much more associated with weird things, like schizophrenia. I don’t want to have that label with me.” (P6) “The first time I had to go shopping (after hospitalization) was horrible. I even took my son with me. I put my head down and I went shopping. I felt very uncomfortable because I made a lot of disturbances in that place.” (P7) “When people asked how I was doing, and it seemed that I was still a little sick, I had to explain why.” (P8) “I always tried to smile, to be active. I worried that people would notice that I was still sick.” (P8) “I was with a boy’s mother to whom I was giving lessons, and in conversation, I told her that I had been hospitalized. She said: - Oh, well, that cannot be. And she decided to cancel the explanations because this could have some opposite effect on the boy... harmful. And I understood, said okay... and the lessons ended.” (P8) “The fact that I was hospitalized and that some people knew was not very favorable for me, especially in my job. In other words, being hospitalized for them was a testament to my eccentricity. It was further proof that maybe other people’s backbiting about me was justified. You see. And that’s what’s complicated.” (P9) “Since I told them I was schizophrenic, they stopped talking like they used to, they moved away from me. If I’m not the one pushing forward, they won’t even talk to me.” (P10) “It is something that is part of my life. Sooner or later people will find out, and I should tell it from my mouth. I’ve always liked to be honest and if I’m not I start to feel bad about myself. I have to tell.” (P10) “With my boyfriend I still felt a little withdrawn whether to tell him or not. […] But I gained courage and shared it on the phone. I was afraid he wouldn’t accept it. But no. He accepted just fine. I felt really good. I was surprised.” (P5) “I even talked to a friend of mine. I could see that he wasn’t doing well. He wanted to give up the course, he had no motivation, a little of what I went through. I had to tell him my story, the reasons that led me to come here.” (P1)
The haunting memories within the self	“It was a terrible shock for me to stay here [in the hospital]. This was a very traumatic experience. I just didn’t come to the psychologist [at the hospital outpatient consultation] because it’s hard for me to come here. Every time I come it reminds me of the month and a half I was here, especially the first few days when I lost my total freedom without doing anything. It was one of the hardest things in my life, so far.” (P10) “The first time I came here, for the outpatient appointment, the memories of the state in which I entered the ward immediately came to my mind. This is where I’ve been down, with such serious problems. It was automatic. To be here, to enter that door. I wasn’t prepared because it was recent, and I felt a little afraid this would happen again, and I would end up here again.” (P1) “I come here every month for the injection. What I don’t like about the injection is having to come here. Because before I looked inside, to the ward, and now I can’t. I go ahead and don’t look inside. Putting the month, I spent there behind, to the fullest. It’s not a place I liked to be, not in the slightest. It was probably one of the most traumatic experiences I’ve had. A very strong and unpleasant experience, but it’s over.” (P6) “I’ve dreamed two or three times that I went back to the psychiatric hospital. And it was a nightmare because for me internment is a person being imprisoned. The person is trapped. It’s normal for a person to come in here and say: I don’t want to be here, this is not for me. So, it didn’t traumatize me, but there is a little mark.” (P2) “When I got home I was nervous, a little withdrawn because: will it happen again? It’s just that the experiences that followed before being hospitalized were all at home.” (P1) “I remember my mother giving me tea. And I drank and always liked tea. But I can’t anymore… Tea! [Face of repulsion]. It was what I drank here [in the hospital]. I was completely numbed, I was all bah… I mean… I could not even chew. Not even chew!” (P6) “I even remember some things I did in the past, which make me sad because I created them and hurt certain people. Thus, it will be a year now, and when I remember, it makes me sad. Sometimes I cry. I celebrated everything here, New Year’s Eve, Christmas and I still feel sequelae of the situation I lived... I still feel depression.” (P7) “The psychotic break itself is a frightening thing. It’s a terrifying thing, really! That’s already a ghost. It’s already hovering there because a person is always subject to it. So, I can never say that I will never be hospitalized again. So that alone is something that gets in the way.” (P12)
From disconnection to the assimilation of the medicated body in the self	“Coming home had two flavors. One was freedom, relief, because I wasn’t expecting it, and two weeks closed [in the hospital] was a bit complicated. But… then managing work and managing things with medication. I couldn’t concentrate. I wanted to solve things and it seemed that my head did not flow.” (P11) “Of course it was a relief to get out of here [from the hospital], of course it was. But I left here full of medication. I was calfing when I left here. Calf, in the sense that I was doped, was not normal in myself, I was heavily medicated. I got fat. I had lip paralysis. Sometimes I couldn’t reason because it was ruining my system. I wasn’t able to do anything.” (P10) “I left here thinking: these guys at the hospital killed me, from the drugs they gave me, and now I’m going to be reborn again. So, it was as if I had died. They shut me down. My brain didn’t work at all. I felt rotten. It was the worst scene ever. I was dead to life. They killed me so I could come back. And so, the word traumatic is the word that best fits. And somehow our organism, our mind, always wants to get around that kind of bad stuff. I was reborn! But brutally and abruptly… and castrating. Literally castrating, because... I couldn’t get an erection, I couldn’t do any of that. So, it was troublesome and incomprehensible. Not being able to think and to know that even our body is not working properly.” (P6) “I felt different, I felt confused. I was not the same person who had left the house before admission. So, there were times when medication inhibited me from performing certain tasks. I was in a vegetative state because it was a transition. Here in the hospital it was one thing, at home it is another. In the hospital, everything was available. I had room and board. And at home, as a father of two, I had daily tasks and made a huge effort to be present, to accompany the tasks.” (P7) “I was on sick leave for a year and three months. I remember it as a big struggle. I was medicated and there were several medication adjustments, always talking to the doctor about my situation.” (P2) “It wasn’t until a year later that the doctor took those pills that made completely twisted me. I continued to take the same injection, which had nothing to do with those pills. With this injection I noticed differences regarding emotional stability, memory, and I also started making music. Finally, I started making music.” (P10) “I am also adapting to the medication because this holds me back. My legs, it hurts my bones, and I never had bone pain. First, I don’t like taking medication and feeling that it is the medication that is causing it. Damn, so now I get an injection every month, and I have to walk feeling this in my legs and bones... this makes me feel a little uncomfortable. But maybe it has to be, right?!” (P3) “A month later [from hospital discharge], I had an appointment, and I told the psychiatrist I didn’t like the medication. It gave me erectile dysfunction, apathy, lack of energy, and torpor. But she insisted that I continued and scheduled another appointment. But I think I no longer showed up at this new appointment because I had already done the weaning with the support of a psychotherapist. But, this last psychotic episode gave me a different perspective. I didn’t like to take psychiatric medications, I didn’t want to at all. At most, I would take an anxiolytic now and then. But now I have some fear.” (P9) “When I went to live with my boyfriend, I felt the need to give up most of the medication. I have different responsibilities, like a house to manage. And I wanted to do the housework, and I felt stoned. And I say to him: - Look, let’s try [leaving the medication during the day]. On the third day, I was off medication, and we started to see results, faster stimulus-response, ability to do things, my body didn’t feel heavy like I wanted to do things and could not.” (P5) “The first medication made me gain 5 kilos on my belly in the first month. I started to feel bad even because the pants stopped fitting. I told the psychiatrist: - As a former sportswoman and dancer to gain weight, no, please! Because I’m not going to feel good about myself if I’m getting fat. [Psychiatrist]: - Ah, but your health comes first. I said: - Okay, but there must be a medication that can have the same effect and I don’t have to put on weight. And she accepted and changed my medication to one that she says I won’t get fat. That was important. If she had not changed the medication, I would probably stop taking the medication.” (P4)
From recognition to overcoming the fragility within the self	“I’m afraid of falling again, going through what I went through. So that this doesn’t happen, I’m going to work on my psychological, on my body, so that it doesn’t happen. School helped me to control my mind more deeply because if we don’t entertain it, it will entertain us. If I don’t have information, if I don’t go on the internet reading about this, reading about that, or at school, or talking to someone about subjects, I feel like I’m losing control of myself again.” (P1) “Right now, I’m taking half a pill at night, which I think is what I still need… Maybe. I’m afraid of relapsing for some reason. I feel the security of having that little dose of medication. At least for now. To be... to be myself.” (P2) “In the more difficult phase I thought about not having the outbreak again. I became more focused on reality. I thought a lot about my children. That’s what I held on to and that I had to have the strength to go forward. I would have to do everything on a mental level to be calm because of my children. It’s holding on to them, and in time, I will improve. To not give up.” (P7) “It scares me to think that I may not be aware of another outbreak or that I am engaging in behaviors that are not good for me and will harm me mentally, physically, and at work. Yes, I’m afraid. Now, in addition to being on medication, I see a psychologist. We are figuring out how to anticipate a little bit what comes before [a crisis].” (P9) “I became much more alert to everything around me. I was a little alienated, and suddenly, I am constantly aware of everything happening around me… my relationship with my parents and the people around me. I am much more alert. And even with me, of analysis with me. It was something I did not think about before: how I felt. And now it feels like I’m in constant evaluation. I think it’s the need to want to show that everything is okay. I think that’s it. That’s why I constantly evaluate myself.” (P11) “This time it was harder to find work because I decided not to work nights. Because that way, I can rest and manage my things better. I can maintain my mental, psychological, and emotional sanity, etc. Doing nights disturbs me because later during the day, I can’t rest as during the night, and I can’t stay awake all night.” (P5) “I was hospitalized because I started to think that the things that happened on television had something to do with my family and me. When I returned home, the television was on, and the program was pretty upsetting because I continued to associate with things I thought before, so I did not really like that part of watching television. Even today, I only watch subscribed television. So, movies and series I choose to watch. I don’t see anything on television. I don’t care about the news. I don’t care about anything. Anything! Unless it’s football (laughs).” (P6) “I somehow assume an autistic behavior towards what I can eventually [perceive], to avoid some kind of thinking.” (P6) “After hospitalization, I changed how I took care of myself, that is, I worked twelve hours a day, sometimes more. Eating was when I had time, and I ate anything. At night I had little sleep. Now I’ve been resting for eight hours a day, eating healthier, I manage my life differently. That’s why the work I didn’t do today I can do tomorrow. Moreover, I already went to the emergency services because of a situation that created much anxiety. I was worried, I felt a little down and needed to come. I am looking at life differently, enjoying those moments I have without doing anything, and trying to make those moments more meaningful.” (P3) “This whole experience was very enriching to understand a lot in life, a lot. It was all part of my personal growth process. I would even say it is a spiritual journey of openness to others and life. Growth is self-knowledge which is knowing that: Well! If conditions appear, I may relapse again, so I must be careful with my life and the conditions I create around me. I feel much stronger now than I was before.” (P2)

**Table 2 jpm-12-01938-t002:** Illustrative quotes for themes regarding the constitutive pole of relationships.

Themes	Illustrative Quotes
The relationship with health professionals: from expectation to response	“When I enter the doctor’s (psychiatrist’s) office, the first thing I want to tell her is all the things I’ve been doing since the last appointment. I want to show the doctor that I’m okay. I also feel this need to show the doctor, who received me, who helped me, that she did a good job. In addition to knowing that her job is well done, she is glad I can move forward with my life. So, if the doctor says to take it, I take it. However, I have been reducing. Until further notice from the doctor that is what I’ll do. It does not confuse me. I would not take medication. I don’t think I need it, but it’s not because I think I will stop taking it. It ends when the doctor thinks it ends. I am not in charge.” (P1) “I have to give myself to doctors and people who understand. I was not used to it, and maybe, in time, medication will be diminished or adjusted when I feel better, when the doctor thinks I’m feeling better. The doctor has to analyze. These diseases cannot be seen. Deep down, I sometimes find it difficult to analyze these psychological things. I don’t understand.” (P3) “The psychiatrist wants to keep the medication, and I have to accept. Okay, if it’s the recommendation. She understands more of the situation than I probably do. I’m discovering it myself.. But… it’s more medication. I don’t think that… I don’t know. I liked to know, I liked to feel, to perceive myself in my natural state to be sure if this medication is necessary or not. Okay, maybe my natural state could, as the doctor said: - Ah, it could trigger another [psychotic outbreak]. Or maybe not! Fifty-fifty. So, now, I have to manage by the fifty that is the demented part. No, I have the other part. I have to try it. Me without medication I have to experience how I deal with things naturally.” (P11) “I was the one who called the doctor’s attention to see if I needed a psychologist to support me. To give some meaning, to justify things because I felt this was out of this world.” (P10) “I needed more support, and my parents preferred that I be followed in the private sector because here, I had a psychologist every month, which is good but not enough. So, I am seeing another psychologist once a week. He helps me vent and talk a little about my fears, concerns...” (P8) “After discharge, I had an appointment a month later. One month! I felt this need to talk to someone neutral to the family, to society. I felt a lack of follow-up, talking about the day-to-day adaptation after hospitalization, doubts about my physical and mental state, and situations that happened outside. Because being hospitalized here also makes us a little more negative, and the medication gives a reaction, and my clinical status after hospitalization was not perceptible to me, nor was it perceptible to know for sure the problem that I had.” (P3) “The psychiatrist was fundamental. We meet twice a year. In the beginning, it was not much more. I feel very confident because he is not someone who gives me the medication: Take, take, take. No. We talk, and whenever I come to Dr. Eduardo we adjust things. I feel an understanding.” (P2) “When I left here [from the internment], I signed up for some psychology consultations. It was something I had before. I’m still learning to gain trust towards her, but I feel that after a conversation, I get answers. I withdraw a confirmation of my thoughts. I feel validated by what’s going on in my head, and I notice I can have certain types of conversations I can’t have outside.” (P11) “It was essential to have found a therapist I like. That is important. I continue to go there. Less times, but I continue.” (P6) “I had the nurse’s support so I could vent. It was fundamental. There were family meetings where we discussed several issues. It created an opening to talk about my problems, giving me the will to talk about what I felt. And even today, it is extremely important to count on that support. Less often but… I feel that there is always that availability.” (P5) “I spoke here in family therapies with nurse Tiago and Isabel, and I vented here. I felt relieved. I felt like I was in a place where I could talk about the disease without being ashamed. Because I felt, I feel social shame. It’s not much, but when I came here, I felt good because we talked about the disease and a little bit of everything. Even on a sexual level. So, everything was part of it. It was a place of comfort. I felt good with their presence. They were very helpful.” (P7)
The relationship with others: reformulating the bonds of alterity	“My parents are more present and concerned about me. My twin sister is always around me, constantly worried about me. And they ask a lot of questions about how I’m doing, how I’m feeling. It brings more security, and stability. I feel better about myself too. And I always need to be with my family to feel safe.” (P8) “The connection again with my mother, my father, and the family made me feel more at ease. It was like bonding again. I already knew them, but it was like reinforcing our bonds. My sister said: - You came back to life. In the sense that they didn’t know me in the way I was, with such serious problems. And when I got home again, I realized: it was really good to have been hospitalized. It helped me to get back to normal.” (P1) “My hospitalization was a shock for my mother and my ex-husband because I think I put myself in the warrior role. I could do anything and endure anything. These were the two relationships that were not going well and that, in a way, were hurting me emotionally. And then they stopped attacking me and started defending me. This, in a way, showed other people that I’m not superwoman. I can also have health problems. I think I had that positioning in life, and now I don’t.” (P4). “So, my relationship with my mother changed profoundly for the better because we were angry and didn’t talk to each other. The approach happened here, while I was hospitalized, with her visits. Let’s say our love grew stronger.” (P4) “My parents became more cautious in the way they treated me. My mother was more… like she was happy because I was there again, dependent on her. During the recovery period, that super-controlled environment was good for me to feel safe and to come back to myself. But afterwards it suffocates me.” (P2) “And all of a sudden, this year it seems like I have my family asking daily how I’m doing, which was something that didn’t happen. Okay, it’s a worry, but worry sometimes has limits and, sometimes, I notice that if I open space, suddenly they are interfering in my life, very easily.” (P11) “I am aware of the harm I have done to my family. I even wrote a letter apologizing because I blamed my mother for the hospitalization. I recognize that if it were not them... And today I am grateful to them.” (P5) “Besides me, my mother suffered a lot from my illness. And today, everything I do is a little bit for her, to show that I’m okay, to feel that she is okay because all this motivation also derives a lot from my mother. I do everything to help her. I started studying to make her proud. When I was sick, I created heartache. I left a sad mark on her, in a negative way. So now I want to mark her positively.” (P1) “When I returned home, I felt I didn’t want to disappoint my parents. I wanted things to go well.” (P8) “I consider that my attitude towards my parents is a little different from what it was before admission. The father-son and mother-son relationship have changed a lot. It has become much more normal. So, I stopped being a child and became a person without being bossed around or shouted at. So, I assume my decisions, my beliefs, I say what I have to say.” (P6) “There was a time when things got a little darker. My wife started to lose her… she believed I could not have a job. Things got the feeling of giving up, of moving away: Maybe the best thing for you is if you stay, treat yourself, and I’ll go on with my life. But it ended up not happening. We are together. We remained friends. The friendship was not lost because she recognized the disease, my diagnosis, and what I was going through. And she grew up. I also grew up in this situation and we are… we can say we are okay. If we are talking about love or that, I don’t know if it exists, but friendship exists. There is friendship.” (P7) “A dear friend who saw me and helped me on one of the most terrible days of my life [the day of the psychotic break] looked at me afterward and said: - Ah, I felt like we could have lost you but no.-You’re back.-Serene.-There you are again. So, I feel like I’m back.” (P4) “With the internment, I became a little less sociable. I haven’t been with my friends or people outside the house. They invite me less to things, I don’t know. Maybe I’m also a little quieter with them. I don’t feel like sharing how I am and how I feel with them.” (P8) “I even cut off relations because my relationships there were unhealthy, so I completely distanced myself from everyone. I disconnected from my life until my internment. I managed to get around with new friends.” (P6) “Although some contacts happened soon after, I cut them off as time progressed. I cut off with friends from this world of mud [drugs] because they consume.” (P5) “I made a friend here at the internment. We talk now and then. We have coffee and talk about the internment, what we felt, what we experienced, the people who were there...ahhh... and that’s good.” (P8) “I became a very good friend with a person hospitalized here at the same time. We were each other’s protectors. We... we became like brothers. It was very complicated for me not to see him after I left because I didn’t know how he was dealing with the situation. But our mothers became very close, and that’s how we got in touch. We started going to each other’s houses, meeting and talking. And then we started a friendship.” (P5)
The relationship with the world: reconnecting as a sense of self	“In the time I was sick, I lost the goals I had for my life. I was lost. I spent my days against the walls, doing nothing. I was a bit disconnected from the world. When I left here [the hospital], I was afraid that it would happen to me again. But then I thought: No, it won’t. If the doctors sent me home, they think I’m OK to start my life. If I’m given a new opportunity to enjoy life, I’ll do everything totally different. I took my life back. I took old goals that I had but with another motivation, with another desire to do them.” (P1) “When I got out of the hospital, it was very difficult. I was very lonely, and I had my parents supporting me but… I felt like I was alone.. It is a very difficult time because during the day people we know, friends, are working and a person during the day has nothing to do. It’s very dull. I was at my own pace, with an unstructured schedule, focused on the house where I lived alone. I worked or stayed up late, slept a lot in the morning, and occupied myself with minor thesis stuff.” (P2) “When the sick leave ended, I decided I had to get out of here because I felt so isolated. I decided to go to another city to try to be more integrated, to try to have a working schedule. I needed to feel the energy to live and the will to live. I decided that I had to change something in my life, change the conditions around me to achieve… Overcome. Gradually I began to have social bonds and a daily routine. It gave me structure, which I lacked.” (P2) “It was good to be back with my things, to be with my belongings, to be in my room… to be comfortable with the surroundings, with everything… But… staying at home for me is a little tiring… It’s boring to stay at home. So, I preferred to be doing something, and after a month I went back to volunteering. When I could, I returned to college classes. This last semester didn’t go very well. I didn’t make any curricular units. But at least I went there, I was active.” (P8) “After [the internment] I didn’t know what to do, and continuing college was not an option. I spent three or four months at home doing nothing. Then, recreationally, I decided to take a multimedia course for four months because I didn’t do anything else. I didn’t know how to do anything! In the end, the teacher called me and told me about the possibility of going to work with him. I accepted. I felt good because it was a way to start doing something related to what I like and because that would bring me some economic benefits, no matter how small, and responsibilities. I felt responsible. I felt useful. After a year or so, I stopped working and decided I wanted to go back to school because several things changed. I wanted to stop smoking tobacco, go to the gym, ride a bike, and do many things. I enrolled in a multimedia graduate degree and just finished the first year, with only one curricular unit to do. Moreover, I had good grades, which is something I never had.” (P6) “My parents decided to enroll me in a geriatrics course to develop new skills because I felt a bit stuck. So… I have gaps in my head. There are areas, there are things I want to put out of there, and I can’t because I don’t remember. I had to make a very big effort to do the tests, to… because the memory, kinda, died, stagnated. And so that was a stimulus to revive things I had in my head and develop new learning skills. When I finished the course, I had a 20% discount on another course. I was already better, and I had better grades. I felt a huge difference.” (P5) “On my initiative, I started studying, leaving the house, and socializing. All shaken, I started to try to move, to have some sense of life that, for me, without a sense of life, it’s not worth being here. Fortunately, I managed to go to the Rec Center, where I started to socialize, go out, see my music friends again, and integrate myself. This helped me a lot, and I didn’t give up and tried to move forward.” (P10)

## Data Availability

Data are held securely by the research team and may be available upon reasonable request and with relevant approvals in place.

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
