# Peer review of "Everyday Life after the First Psychiatric Admission: A Portuguese Phenomenological Research"

_jpm, 2022, doi:10.3390/jpm12111938_

Round 1

Reviewer 1 Report

Dear authors, the experiences reported show the proposed perception, which (un)reveals a strong influence of stigma of social discrimination exercised on the person under psychiatric treatment.

With regard to nursing care, the instrument can serve and constitute a basis for the establishment of transitional care for psychiatric patients.

I emphasize the importance of the material for the academic community.

Sincerely.

Author Response

Dear Reviewer 1

We sincerely thank you for the comments and the recognition of the importance of this paper. We have place the changes to the paper bellow, hopefully easing the article's readability. We intend to pursue this matter concerning future research, and your comments were genuinely encouraging.

Kind regards
The authors

Changes to the reviewed paper:

- Regarding the introduction, we strove to reorganize and clarify the sequence of conceptual though that supports the need for this study. The changes are highlighted in red. The research question is underlined in green and no hypotheses can be made since this is a phenomenological study which seeks to discern the primordial meaning embedded in lived experience. According to Van Manen, the researcher has to set aside is previous knowledge and assumptions, in a movement called the epoché, that intends to remove any obstruction which could digress the ability to achieve a more original and inchoative contact with the lifeworld.

- The chosen method – phenomenology of practice – is congruent to the research question and aims as it provides the means to access the lived experience of this particular phenomenon. In the section of Materials and Methods we sought to precisely describe the underpinnings of the method, as well as the concrete steps ensued in order to achieve the goals stipulated for this study. Even though its disposition may not correspond to the usual form seem in research articles (e.g. Participant Recruitment and Selection; Data Collection; Data Analysis), its sequence is faithful to phenomenology of practice, provides a detailed description of the steps and decisions made, and allows for a broader perspective relatively to the way researchers may describe their research methods.

- The results are now from page 6-16 with essential statements of the interviewed participants summarized in two tables. Great care was ensued has we tried to preserve the vocative value of participants’ narratives.

- Regarding the discussion some improvements were made regarding our findings and its relation to previous literature.

Reviewer 2 Report

The authors submitted a paper concerning the topic “Everyday life after the first psychiatric admission: a Portuguese phenomenological research”
(jpm-1910484).

The topic is important however the paper must be carefully revised.

There is no clear Introduction, no clear research question, and no hypotheses.

 It will be better to reorganize the introduction and make sure each paragraph combines things that are related together.

Page 6-32:

It is important to shorten the results to the essential core statements. The results are not acceptable in this length. Essential statements of the interviewed patients must be summarized in a table. This makes the paper also easier to read.

There should less restatement of the findings and a much more thorough discussion of what the findings mean in relation to each other and in relation to the previous literature.

Author Response

Dear Reviewer 2,

We thank you for your analysis and feedback that have allowed us to improve the paper.

With regards to the issues you point out we would like to reply:

- Regarding the introduction, we strove to reorganize and clarify the sequence of conceptual though that supports the need for this study. The changes are highlighted in red. The research question is underlined in green and no hypotheses can be made since this is a phenomenological study which seeks to discern the primordial meaning embedded in lived experience. According to Van Manen, the researcher has to set aside is previous knowledge and assumptions, in a movement called the epoché, that intends to remove any obstruction which could digress the ability to achieve a more original and inchoative contact with the lifeworld.

- The chosen method – phenomenology of practice – is congruent to the research question and aims as it provides the means to access the lived experience of this particular phenomenon. In the section of Materials and Methods we sought to precisely describe the underpinnings of the method, as well as the concrete steps ensued in order to achieve the goals stipulated for this study. Even though its disposition may not correspond to the usual form seem in research articles (e.g. Participant Recruitment and Selection; Data Collection; Data Analysis), its sequence is faithful to phenomenology of practice, provides a detailed description of the steps and decisions made, and allows for a broader perspective relatively to the way researchers may describe their research methods.

- The results are now from page 6-16 with essential statements of the interviewed participants summarized in two tables. Great care was ensued has we tried to preserve the vocative value of participants’ narratives.

- Regarding the discussion some improvements were made regarding our findings and its relation to previous literature.

Once again, we appreciate you feedback and we remain willing to make the changes you may find crucial.

Kind regards,

The authors

Reviewer 3 Report

The authors submitted a paper concerning the topic “Everyday life after the first psychiatric admission: a Portuguese phenomenological research”
(jpm-1910484).

The paper must be carefully revised.

There is no clear Introduction, no clear research question, and no hypotheses.

 It will be better to reorganize the introduction and make sure each paragraph combines things that are related together.

Page 6-32:

It is important to shorten the results to the essential core statements. The results are not acceptable in this length. Essential statements of the interviewed patients must be summarized in a table. this makes the paper easier to read.

There should less restatement of the findings and a much more thorough discussion of what the findings mean in relation to each other and in relation to the previous literature.

Author Response

Dear Reviewer 3,

We thank you for your analysis and feedback that have allowed us to improve the paper.

With regards to the issues you point out we would like to reply:

- Regarding the introduction, we strove to reorganize and clarify the sequence of conceptual though that supports the need for this study. The changes are highlighted in red. The research question is underlined in green and no hypotheses can be made since this is a phenomenological study which seeks to discern the primordial meaning embedded in lived experience. According to Van Manen, the researcher has to set aside is previous knowledge and assumptions, in a movement called the epoché, that intends to remove any obstruction which could digress the ability to achieve a more original and inchoative contact with the lifeworld.

- The chosen method – phenomenology of practice – is congruent to the research question and aims as it provides the means to access the lived experience of this particular phenomenon. In the section of Materials and Methods we sought to precisely describe the underpinnings of the method, as well as the concrete steps ensued in order to achieve the goals stipulated for this study. Even though its disposition may not correspond to the usual form seem in research articles (e.g. Participant Recruitment and Selection; Data Collection; Data Analysis), its sequence is faithful to phenomenology of practice, provides a detailed description of the steps and decisions made, and allows for a broader perspective relatively to the way researchers may describe their research methods.

- The results are now from page 6-16 with essential statements of the interviewed participants summarized in two tables. Great care was ensued has we tried to preserve the vocative value of participants’ narratives.

- Regarding the discussion some improvements were made regarding our findings and its relation to previous literature.

Once again, we appreciate you feedback and we remain willing to make the changes you may find crucial.

Round 2

Reviewer 2 Report

Thank you for taking the time to improve your manuscript. After reading the additions as suggested by all reviewers, I have nothing more to add or suggest, and I agree this version of the paper to be published.

With regards,

reviewer